# Altered mucins and aquaporins indicate dry eye outcome in patients undergoing Vitreo-retinal surgery

Ramalingam Mani[1], P. S. Shobha[2], Saravanan Thilagavathi[2], Padmanabhan Prema[3], Natarajan Viswanathan[4], Ratra Vineet[5], Ratra Dhanashree[6]*, Narayanasamy Angayarkanni[1]*

1 RS Mehta Jain Department of Biochemistry and Cell Biology, Vision Research Foundation, SankaraNethralaya, Chennai, India, 2 Elite School of Optometry, Medical Research Foundation, SankaraNethralaya, Chennai, India, 3 Department of Cornea and Refractive Surgery, Medical Research Foundation, SankaraNethralaya, Chennai, India, 4 Department of Bio-Statistics, Vision Research Foundation, SankaraNethralaya, Chennai, India, 5 Department of Comprehensive Ophthalmology, Medical Research Foundation, SankaraNethralaya, Chennai, India, 6 Department of Vitreo-retinal Diseases, Medical Research Foundation, SankaraNethralaya, Chennai, India

* drak@snmail.org (RD); drdad@snmail.org (NA)

**Data Availability Statement:** All relevant data are within the manuscript and its Supporting Information files.

## Abstract

Vitreo-retinal (VR) surgeries induce conjunctival changes. However, there are no study reports regarding prevalence and severity of dry eye after these surgeries. This study evaluated dry eye outcome after VR surgery. Patients undergoing VR surgery classified as scleral buckle and microincision vitrectomy surgery (n = 44, mean age: 56.09±10.2 years) were recruited. Dry eye evaluation was done before and 8 weeks after surgery (2 weeks after omitting topical eye drops). Conjunctival imprint cytology for goblet cell count and tear Mucin 5AC (MUC5AC) protein estimation was done. Gene expressions of MUC5AC, MUC4, MUC16, Aquaporin 4 (AQP4) and AQP5 were analyzed in the conjunctival imprint cells by qPCR. None of the patients exhibited clinical signs of dry eye after VR surgery. But the conjunctival goblet cell density (GCD) was significantly lowered post-VR surgery (63% cases, **p = 0.012) with no alterations in the tear MUC5AC protein. Post-VR surgery, the conjunctival cell gene expression of MUC4, MUC16 and AQP4 were significantly increased (*p = 0.025, *p = 0.05 and *p = 0.02 respectively) and AQP5 was significantly lowered (*p = 0.037), with no change in MUC5AC expression. Tear cytokines were significantly increased post-VR surgery (anti-inflammatory: IL1RA, IL4, IL5, IL9, FGF; PDGFbb and pro-inflammatory: IL2, IL6, IL15, GMCSF and IFNg). Though clinical signs of dry eye were not observed after VR surgery, ocular surface changes in the form of reduced GCD, altered MUC5AC, MUC4, MUC16, AQP4, AQP5 and cytokines are suggestive of dry eye outcome at the molecular level especially inpatients aged above 51 years, especially female gender and those who are diabetic.

**Funding:** This work was supported by Novartis Healthcare Private Limited. The funder had no role in study design, data collection and analysis, decision to publish, or preparation of the manuscript.

**Competing interests:** The authors have read the journal's policy and have the following conflicts: Novartis Healthcare Private Limited funded this project. This does not alter our adherence to all the PLOS ONE policies on sharing data and materials.

## Introduction

Dry eye syndrome (DES), a common ophthalmological problem, resulting from either excessive tear film evaporation or decrease in the tear production, can affect the quality of life. Increasing DES frequency has been noted with increasing number of office workers using visual display terminals, and increased exposure to various industrial agents, medications and cigarette smoke. Disturbances of the ocular tear film are common complications following ocular surgeries. DES in addition to causing various disabling symptoms has been shown to influence the outcome of cataract and corneal surgical procedures [1].

Pars plana vitrectomy (PPV) is the treatment of choice for many vitreo-retinal disorders including retinal detachment, vitreous hemorrhage, advanced proliferative diabetic retinopathy, macular hole and others. During PPV, conjunctiva is traumatized by peritomy and by scleral depression [2]. Morphological changes of the conjunctiva and distributional changes in ocular mucins were observed after this type of surgery [3–4]. Over the past few years, PPV has been revolutionized by the development of small gauge transconjunctival sutureless technique. This technique can likely cause less dry eye due to fewer traumas to the conjunctiva. Wasfy et al [5] evaluated the changes in the tear film quality following 20G and 23G PPV but found no significant differences between the two groups. However, the frequency and severity of dry eye symptoms in patients following such small gauge VR surgery has not been addressed objectively in the previous literature. This study explores the dry eye status before and after VR surgery based on clinical evaluation and changes at molecular levels in terms of tear fluid cytokines, mucins and aquaporins.

## Materials and methods

The study was conducted as a hospital based prospective observational study after institution ethics committee approval ("*Ethics Sub-Committee (Institutional Review Board) No*: *573-2016-P Title*: *Prevalence of dry eye following retinal surgery*"). Written and informed consent of the patients was obtained. Consecutive patients, ≥40 years of age undergoing VR surgical interventions were recruited in the study. The VR surgeries included scleral buckling procedure which required 360˚ conjunctival opening and transconjunctival micro incision vitrectomy surgery (MIVS) which did not require any conjunctival opening. Dry eye evaluation based on Dry Eye Workshop (DEWS) [6] criteria 2007 was done to rule out any preexisting DES. The tear samples and imprints were collected before and after VR surgery. Pre-surgery samples were collected 3–5 days before the surgery. A standard protocol for surgery and postoperative care were followed in all the patients. After the surgery, the patients were asked to use topical prednisolone acetate eye drops starting with 6 times a day and tapered over 6 weeks along with atropine eye drops 2 times a day for 2 weeks. Two months after the surgery, the patients were re-evaluated for DES. Two weeks prior to re-evaluation, the patients were asked to stop all the topical medications prescribed after the surgery. The exclusion criteria were previous diagnosis/history of dry eye, use of any tear supplements or lubricants, contact lens wear, history of acute or chronic ocular inflammation, history of any intraocular surgery including refractive laser surgery in the past, and pregnancy status.

The study patients were evaluated for symptoms of dry eye with McMonnies questionnaire (Appendix 1). Clinical evaluation of DES was done by Schirmer's test, tear osmolarity (TOSM) test, tear film break up time (TBUT), and tear meniscus height (TMH), Fluorescein staining of the conjunctiva was done. The molecular evaluation of dry eye was done based on tear analysis of cytokines, MUC5AC protein concentration and imprint cytology for conjunctival gene expression of mucins, aquaporins.

## Tear collection and protein extraction

Tear samples were collected by commercially available pre-sterilized Schirmer's strips (Conta care, Baroda, India). The strips of 5x35 mm size were placed in the lower conjunctival cul-de-sac for 5 min with the eye closed and without any topical anesthesia. A wetting of ≤10mm was taken as an indicator of dry eye. The Schirmer's strip was then kept in a sterile vial, and transported in an ice box to the laboratory within 30 min. They were stored at -80˚C until further processing.

The tear-soaked strip was placed into a 1.5 ml vial to which was added 250 μl of 1X phosphate buffered saline with 0.1% Triton X-100 and protease inhibitor. The vials were centrifuged at 8000 rpm for 10 min at 4˚C. The supernatant was collected and pellet was removed. The extracted tear sample total protein concentrations were estimated by BCA assay method [7]. Protein assay reagent kit (BCA Protein Assay Kit; Pierce) was used to determine the protein concentration in the tear samples.

## Assay for MUC5AC concentration in tears by ELISA

The concentration of the secreted MUC5AC in the tear samples were quantified by Enzyme-Linked Immuno Sorbent Assay (ELISA)(SEA756Hu; Cloud-Clone Corp, USA). All samples were analyzed according to the manufacturer's guidelines. Absorbance was measured at 450nm. The MUC5AC concentration was normalized to the tear protein content and expressed as MUC5AC protein (ng) per tear total protein (mg) [8].

## Cytokines profiling

A Bio-Plex multiplex assay kit (Bio-Plex Human Cytokine 27-plexpanel, Bio-Rad Laboratories, Hercules, California, USA) was used to measure the concentrations of 27 cytokines in the tears, before and after VR surgery. The analysis was performed according to the manufacturer's instructions and read in Bio-Plex Reader (Bio-Rad Laboratories). Tear cytokines were detected using multiplex bead assay. Standard curves were generated using the Bio-Plex Manager System (software V.6.0; Bio-Rad Laboratories) to calculate the cytokine concentrations in the tear samples. Cytokines: (Interleukin (IL)-1ra, IL-1b, IL-2, IL-4, IL-5, IL-6, IL-7, IL-9, IL-10, IL-12 (p70), IL-13, IL-15, IL-17, granulocyte-macrophage colony stimulating factor (GM-CSF), granulocyte- colony stimulating factor (G-CSF), interferon-γ (IFN-γ), tumor necrosis factor (TNF)-α)), Chemokines: (IL8, eotaxin (CCL11), interferon inducible protein-10 (IP-10)/CXCL10, monocyte chemo attractant protein (MCP)-1/CCL2, macrophage inflammatory protein-1a (MIP-1a)/CCL3, MIP-1b/CCL4, regulated upon activation, normal T cell expressed and presumably secreted(RANTES)/CCL5) and Growth factors: (Fibroblast growth factor (FGF), platelet derived growth factor (PDGF)-BB, vascular endothelial growth factor (VEGF). The cytokine concentrations were expressed as pg/ml [9].

## Collection of conjunctival impression cytology (CIC)

Collection of cell imprints from the superior, inferior, nasal and temporal regions of the eye was done after topical anesthesia with 0.5% proparacaine hydrochloride (Sunways India Pvt Ltd). Sterilized hydrophilic PVDF (H-PVDF, size: 5x5 mm$^2$) membrane filter paper was pressed gently against the temporal bulbar conjunctiva adjacent to the corneal limbus, with blunt, smooth-tipped forceps for 5-10 seconds, and removed. The imprint was stored at -80˚C until further processing [10]. These imprints were used for gene expression analysis namely, MUC5AC, MUC4, MUC16, AQP4 and AQP5 and mucin secreting goblet cells staining.

## Periodic acid and Schiff's reagent (PAS) staining

The conjunctival imprint was stained using periodic acid and Schiff's reagent (Hi-media, India) for goblet cells detection. Zhang et al [11] protocol was followed with modifications. The imprint was fixed with 95% ethanol for 5 min and washed with distilled water. Cells were incubated with 0.5% periodic acid solution for 10 min, rinsed with tap-water, stained with schiff's reagent (1:1 ratio) for 10 min, followed by rinsing in water. Finally, counterstained with haemotoxylin solution (Hi-media, India) for 30 seconds and washed in tap-water followed by distilled water, dried and mounted using DPX mountant. All the steps were performed at room temperature. Goblet cells were counted using Inverted microscopy (Nikon Eclipse Ts2, USA). The conjunctival epithelial cells were stained blue and the goblet cells were stained pink. Total numbers of goblet cells were counted on the whole imprints surface area ($5x5 \ mm^2$). The number of goblet cells was recorded both before and after VR surgery.

## RNA isolation and quantitative PCR (QPCR)

Total RNA was isolated from the conjunctival imprint samples using Tri-reagent (InvivoGen, San Diego, California, USA) according to manufacturer's protocol. Total RNA was converted to complementary DNA (cDNA) synthesis using iScript cDNA synthesis kit (Bio-Rad Laboratories) as described in the manufacturer's protocol. The gene-specific primers were used for the genes: 18S rRNA: `F-AACCCGTTGAACCCCATT`, `R-CCATCCAATCGGTAGTAGCG` (product size: 149bp); MUC5AC: `F-AATGGTGGAGATTTTGACAC`, `R-TTCTTGTTCAGGCAAAT CAG` (173bp); MUC4: `F-AGATTTTCTCCTACCCCAAC`, `R-TTCCTGATAAAATGTGGTCC` (125bp), MUC16: `F-TAAAGGACTACACACAGGAG`, `R-AAAGTGGGAGTATAGACACTG` (126bp), AQP4:`F-CATTGGATATATTGGGTTGGG`, `R-TCAACATCTGGACAGAAGAC` (83bp) and AQP5:`F-CTTGTCGGAATCTACTTCAC`, `R-AGCAGGTAGAAGTAAAGGATG` (155bp). Real-time PCR was performed with the light cycler 96 (Roche Diagnostics, Switzerland) using the SYBRGreen chemistry. The comparative $2(-\Delta Ct)$ method was used to analyze the results of the genes of interest relative to internal control gene (18rRNA) [9].

## Protein–protein interaction network analysis

Protein-protein interaction network analysis was done (12[th],September,2018) for significantly altered parameters using UNIPROT and STRING databases (http://string-db.org/,version 10.5) [12] to highlight functional enrichments using a number of functional classification systems such as Protein families (Pfam) and Kyoto Encyclopedia of Genes and Genomes (KEGG) (version 10.5) [13].

## Statistics

Data were expressed as mean±SD along with the median and range. All statistical analyses were performed using SPSS.14.0. Student's t-test and the non-parametric Mann-Whitney U test were used to compare the means of the quantitative variables between two independent groups. The Kruskal-Wallis test was used to compare multiple groups. One-way analysis of variance (ANOVA) was used to assess the correlation between pre and post-surgery sample and within parameters. A p-value <0.05 was accepted as statistically significant. Descriptive statistics was computed for all the continuous variables for these three groups. Since the sample size for the three sub groups are small and do not satisfy the normality assumption, we performed non parametric test.

## Results

This study was performed between March-2017 to April-2018. It included 44 patients with 31 (70%) males and 13 (30%) females. The overall mean age was 56.09± 10.2 years. The age of the males was 55.94 ± 9.89 years and in females it was 56.46 ± 11.3 years.

### Clinical evaluation of DES pre and post surgery: Lowered goblet cell count

The clinical evaluation of dry eye pre and post-VR surgery revealed no significant dry eye development based on the dry eye work up done to evaluate DES (Table 1).

For further analysis based on type of surgical insult and presence of vitreous hemorrhage, the eyes were divided into various subgroups, namely those which required 360˚ conjunctival opening, versus those where the conjunctival was not opened; eyes which required sutures through the conjunctival versus eyes which were non-sutured; eyes with vitreous hemorrhage versus, no vitreous hemorrhage. Mann Whitney U test was used to test the difference in median values between the two groups. Clinically, none of the groups showed any significant dry eye symptoms after VR surgery. The clinical tests as enlisted in Table 1 did not show significant differences when compared between the groups.

However, the conjunctival GCD done by imprint cytology was significantly lowered after VR surgery compared to pre-surgery (p = 0.012). The distribution analysis revealed that 63% of cases had lowered GCD after VR surgery when compared to the pre -surgery levels in terms of the median value (Fig 1a).

### Conjunctival mucins and aquaporins gene expressions are altered post VR surgery

Goblet cells are the major source of tear mucins. Mucins are functional with aquaporins to promote the gel forming mucin. Conjunctival imprints were analyzed for mucin and aquaporin gene expression pre and post VR surgery. MUC4 and MUC16 gene expression were significantly up-regulated (p = 0.025 and p = 0.05 respectively), while MUC5AC gene expression was not significantly altered post-VR surgery (Fig 1b). AQP5 gene expression was significantly down-regulated (p = 0.037), while AQP4 gene expression was up-regulated (p = 0.02)post-VR surgery (Fig 1c).

GCD and AQP5 genes expression levels were significantly lowered post-surgery in the subgroups based on surgical intervention, non-suture; conjunctival non-opening and vitreous hemorrhage (Table 2). MUC5AC though lowered was not significant in the same conditions. There was no difference prior to VR surgery amongst the groups.

**Table 1. Clinical parameters evaluating dry eye pre and post vitreo-retinal surgery.**

| S.No | DES parameters | Pre-surgery | | Post-surgery | | p-value |
|---|---|---|---|---|---|---|
| | | (n = 44) | | (n = 44) | | |
| | | Mean ± SD | Median | Mean ± SD | Median | |
| 1 | Schirmer's (mm) | 12.2 ± 7.78 | 11.5 | 14.3 ± 10.63 | 12 | 0.291 |
| 2 | Tear Osmolarity (mOsm/L) | 305.8 ± 18.01 | 304 | 300 ± 11.9 | 300 | 0.08 |
| 3 | Tear meniscus height (mm) | 0.28 ± 0.11 | 0.27 | 0.3 ± 0.1 | 0.28 | 0.414 |
| 4 | Tear break up time (Sec) | 7.34 ± 6.65 | 4.4 | 6.91 ± 5.68 | 4.88 | 0.751 |
| 5 | Corneal and conjunctival stain score | 1.3 ± 1.13 | 1.0 | 1.02 ± 1.24 | 1.0 | 0.291 |

p-value is comparison between pre and post vitreo-retinal surgery of the cases recruited. DES: Dry eye syndrome, mOsm/L: Milliosmole/Litre, Sec: Seconds, mm: millimeter.

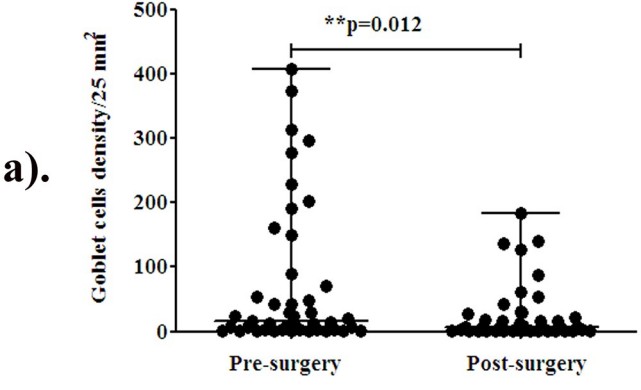

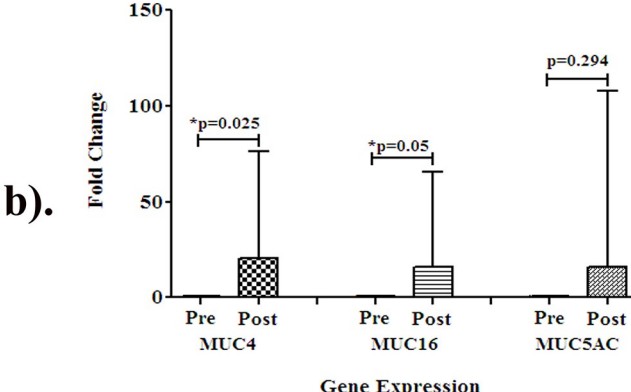

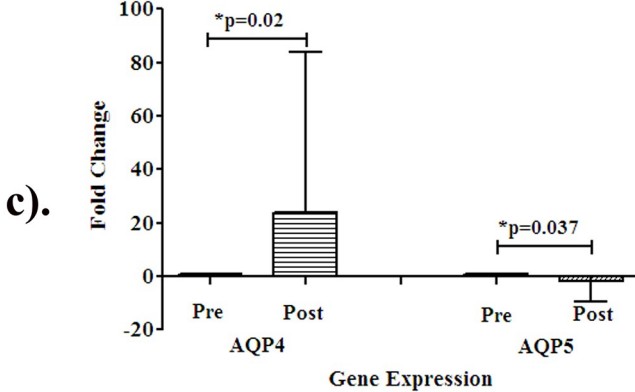

**Fig 1. Distribution analysis of conjunctival goblet cells and gene expression analysis of mucin and aquaporins in pre and post vitreo-retinal surgery. a)** Distribution analysis of conjunctival goblet cells. Median value of pre-surgery is 15.5 and post-surgery is 6. **b)** MUC4, MUC16, MUC5AC and c). AQP4 and AQP5 gene expressions in the conjunctival epithelium pre and post VR surgery. Sample size (n = 44); p-value is comparison between pre and post VR surgery. $^*p < 0.05$, $^{**}p < 0.01$.

## MUC5AC concentration in tears

Tear MUC5AC concentration was not significantly altered post-VR surgery (1.52 ± 1.15 ng/ mg of total protein) compared to pre-surgery condition (1.39 ± 0.88 ng/mg of total protein).

Table 2. Changes in goblet cell density, MUC5AC protein and AQP5 gene expressions, pre and post vitreo-retinal surgery.

| DES parameter | VR Surgery status | Non suture (Mean ± SD) | Suture (Mean ± SD) | 360˚Conjunctival opening (Mean ± SD) | Conjunctival non-opening (Mean ± SD) | Vitreous Hemorrhage (Mean ± SD) | Non vitreous Hemorrhage (Mean ± SD) |
|---|---|---|---|---|---|---|---|
| | | (n = 17) | (n = 27) | (n = 3) | (n = 41) | (n = 23) | (n = 21) |
| Conjunctival GCD (25 mm$^2$) | Pre | 87.3±121.63 | 82.74±102.69 | 68.33±106.23 | 73.0±113.87 | 85 ± 118.96 | 59.29 ± 105.53 |
| | Post | 24 ± 27.77 | 41.11 ± 72.8 | 67.68±101.03 | 27.634±58.36 | 18.17± 38.38 | 43.71 ± 77.96 |
| | p-value | *0.044 | 0.092 | 0.994 | *0.026 | **0.014 | 0.59 |
| Tear MUC5AC (ng/ mg of total proteins) | Pre | 1.36 ± 1.0 | 1.62 ± 1.24 | 3.14 ± 1.69 | 1.40 ± 1.03 | 1.38 ± 1.08 | 1.68 ± 1.23 |
| | Post | 1.34 ± 0.98 | 1.42 ± 0.84 | 1.64 ± 0.98 | 1.37 ± 0.89 | 1.37 ± 0.92 | 1.41 ± 0.86 |
| | p-value | 0.93 | 0.5 | 0.25 | 0.89 | 0.99 | 0.42 |
| Conjunctival AQP5 gene expression | Pre | 1 ± 0 | 1 ± 0 | 1 ± 0 | 1 ± 0 | 1 ± 0 | 1 ± 0 |
| | Post | -6.759±13.15 | 0.128 ± 6.505 | 0.66 ± 1.143 | -2.766±10.40 | -0.871 ± 4.26 | -4.353 ± 13.84 |
| | p-value | *0.021 | 0.489 | 0.634 | *0.023 | *0.041 | 0.084 |

p-value is comparison between pre and post VR surgery. DES: Dry eye syndrome, GCD: Conjunctival goblet cell density, MUC: Mucin, AQP: Aquaporin, VR-Vitreo-retinal,

*p<0.05,

**p<0.01.

Tear MUC5AC concentrations post-surgery was decreased in 57% cases and increased in 39% (≥1.2 fold).

## Tear cytokine response post VR surgery

Significant changes in tear cytokine levels were observed post-surgery compared to pre-surgery (S1 Table). Amongst 27 analyzed, six anti-inflammatory cytokines namely, IL1RA (p = 0.046), IL4 (p = 0.008), IL5 (p = 0.027), IL9 (p = 0.019), FGF (p = 0.05) and PDGFbb (p = 0.05) levels were significantly increased. Five pro-inflammatory cytokines namely, IL2 (p = 0.05), IL6 (p = 0.044), IL15 (p = 0.01), GM-CSF (p = 0.016) and IFNg (p = 0.029) levels were significantly increased after VR surgeries. These were altered by more than 2-fold in more than 50% cases. Distribution analysis of significantly altered tear cytokines expression in post-VR surgery cases compared with pre-surgery revealed significant changes in more than 50% cases who underwent the VR surgery with respect to 11 cytokines amongst the 27 tested namely, IL1RA, IL4, IL5, IL9, FGF, PDGFbb, IL2, IL6, IL15, GM-CSF and IFNg (Fig 2). Tear cytokines analyzed based on VR surgeries showed increased cytokine response characteristically in vitreous hemorrhage, non-suture and conjunctival non-opening groups (S2 Table).

## Age, sex and diabetic status influenced the DES outcome in VR surgeries

Conjunctival GCD was significantly decreased in the DM cases post- surgery (Pre: 51.46 ± 88.08 cells/25mm$^2$; Post: 13.5 ± 19.28 cells/25mm$^2$, p = 0.045). It was significantly lower than non-diabetic cases (p = 0.044) (S1 Fig). Conjunctival GCD was not significantly altered in the non-diabetic cases post-VR surgery. Thus, the study revealed that age ≥50 year, female gender and diabetic status influenced the conjunctival GCD, with a net decrease in number thereby predisposing to a dry eye outcome post VR surgery However. Age, gender and presence of diabetes did not significantly alter the tear MUC5AC concentration after VR surgery (Table 3).

Table 4 shows that in non-DM cases there was a significant increase in MUC5AC (p = 0.025); MUC4 (p = 0.027) which was not observed in the DM cases, indicating mucin

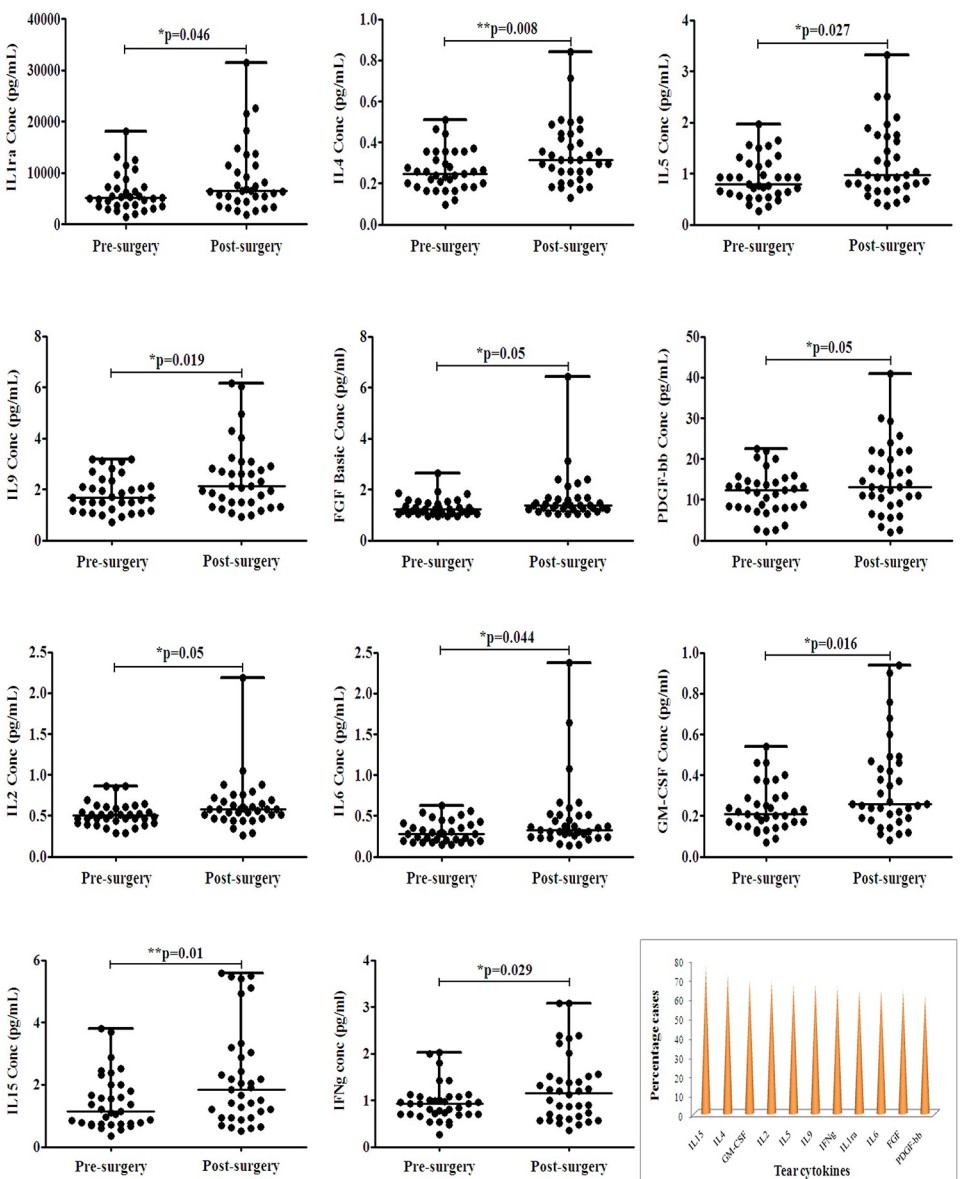

**Fig 2. Distribution analysis of tear cytokines in pre and post VR surgery.** Distribution analysis of cytokine expression in post-VR surgery cases compared to pre-surgery showing the median. The last figure shows a bar graph depicting percentage of cases with significantly up-regulated tear cytokines post VR surgery. Sample size = 36. P-value is comparison between pre and post VR surgery. *p<0.05, **p<0.01.

protection in non-DM cases. In addition, AQP5 is significantly reduced in DM cases (p = 0.04). AQP5 was also found to be increased significantly in males compared to females (p = 0.018). AQP4 increase was found to be high in age ≥ 51 years (p = 0.029) indicating its differential role from that of AQP5.

In DM cases, seven cytokines were significantly up-regulated post-VR surgery that include, IL4 (p = 0.019), IL5 (p = 0.05), IL9 (p = 0.028), FGF basic (p = 0.019) that are anti-inflammatory and IL15 (p = 0.048), GMCSF (p = 0.032), IFNg (p = 0.048), which are pro-inflammatory cytokines (Fig 3a). The distribution analysis of tear cytokines revealed increase in IL15 (77% cases), followed by IL9 (73%), IL4 (68%), IL5 (68%), FGF Basic (68%), GMCSF (63%), and

**Table 3. MUC5AC concentration and goblet cell density in tear samples pre and post vitreo-retinal surgery sub-grouped based on Diabetes Mellitus status, age and gender.**

| S. No | Parameters | Tear MUC5AC (ng/mg of total protein) | | p-value | Conjunctival Goblet cell density (25mm$^2$) | | p-value |
|---|---|---|---|---|---|---|---|
| | | pre-surgery (Mean ± SD) | post-surgery (Mean ± SD) | | pre-surgery (Mean ± SD) | post-surgery (Mean ± SD) | |
| 1 | **Total number of cases (n = 44)** | 1.52 ± 1.15 | 1.39 ± 0.88 | 0.55 | 72.73 ±112.2 | 25.95 ±43.37 | **0.012 |
| 2 | **DM cases (n = 24)** | 1.39 ± 0.82 | 1.31 ± 0.88 | 0.737 | 51.46 ±88.08 | 13.5 ±19.28 | *0.045 |
| 3 | **Non-DM cases (n = 20)** | 1.67 ± 1.46 | 1.49 ± 0.9 | 0.629 | 98.25 ±133.57 | 50.6 ±85.09 | 0.186 |
| 4 | **Age ≤50 (n = 15)** | 1.75 ± 1.39 | 1.56 ± 0.96 | 0.614 | 87.53 ±109.66 | 33.73 ±57.29 | 0.103 |
| 5 | **Age ≥51 (n = 29)** | 1.33 ± 0.89 | 1.25 ± 0.81 | 0.751 | 77.31 ±111.34 | 28.62 ±64.07 | *0.04 |
| 6 | **Male cases (n = 31)** | 1.39 ± 1.2 | 1.39 ± 0.96 | 0.1 | 53.29 ±89.13 | 38.45 ±71.26 | 0.472 |
| 7 | **Female cases (n = 13)** | 1.83 ± 0.99 | 1.4 ± 0.7 | 0.205 | 119.08 ±148.15 | 11.08 ±12.67 | *0.015 |

p-value is comparison between pre and post VR surgery of that respective parameter.

*p<0.05,

**p<0.01.

IFNg (59%). In non-DM cases, there was no significant difference in the cytokine levels in pre and post-surgery (S2 Fig).

Tear cytokines were analyzed in age group above and below 50 years. Age ≥ 51years showed significantly increased levels of IL4 (p = 0.012), IL5 (p = 0.041), IL9 (p = 0.018), FGF (p = 0.05), MIP1a (p = 0.007), MIP1b (p = 0.018) of anti-inflammatory nature and IL6 (p = 0.05), IL15 (p = 0.031), GMCSF (p = 0.01), TNFa (p = 0.032) as pro-inflammatory cytokines (Fig 3b). MIP1a tear cytokine shows higher level (Fig 3c). IL9 (p = 0.047) and IL15 (p = 0.035) tear cytokines were significantly up-regulated in the males.

## Protein-protein interactions network analysis

Protein-protein interaction network analysis was done for significantly altered parameters using UNIPROT and STRING databases. After VR surgery, we found that 18 parameters were significantly altered out of 38 parameters studied (GCD, Mucins, AQPs and cytokines), that included the cytokines, IL1RA, IL2, IL4, IL5, IL6, IL9, IL15, FGF, GMCSF, PDGFbb, IFNg,

**Table 4. Alterations in conjunctival gene expressions of mucins and aquaporins post vitreo-retinal surgery.**

| S. No | Parameters | MUC5AC Fold change (p-value) | MUC4 Fold change (p-value) | MUC16 Fold change (p-value) | AQP4 Fold change (p-value) | AQP5 Fold change (p-value) |
|---|---|---|---|---|---|---|
| 1 | Total cases (n = 44) | 3.98 ± 119.91 (0.294) | 20.29 ± 56.09 (*0.025) | 15.85 ± 49.97 (*0.052) | 24.08 ± 59.79 (*0.02) | -1.71 ± 7.64 (*0.037) |
| 2 | DM cases (n = 24) | -7.45 ± 160.43 (0.798) | 13.83 ± 59.11 (0.293) | 13.53 ± 48.68 (0.214) | 25.76 ± 62.1 (0.068) | -1.64 ± 5.42 (*0.041) |
| 3 | Non-DM E(n = 20) | 17.69 ± 32.01 (*0.025) | 28.04 ± 52.67 (*0.027) | 18.64 ± 52.59 (0.142) | 21.77 ± 58.39 (0.165) | -1.78 ± 9.72 (0.256) |
| 4 | Age ≤50 (n = 15) | 36.55 ± 122.41 (0.27) | 30.17 ± 78.17 (0.16) | 13.65 ± 48.04 (0.317) | 12.29 ± 49.17 (0.416) | -1.11 ± 4.77 (0.14) |
| 5 | Age ≥51 (n = 29) | -12.87 ± 117.16 (0.526) | 15.18 ± 41.2 (0.069) | 17 ± 51.73 (0.101) | 30.21 ± 64.71 (*0.029) | -2.01 ± 8.81 (0.101) |
| 6 | Males (n = 31) | -15.95 ± 111.76 (0.402) | 9.75 ± 33.58 (0.152) | 11.13 ± 49.52 (0.259) | 17.32 ± 55.26 (0.138) | -1.85 ± 5.92 (*0.018) |
| 7 | Females (n = 13) | 51.5 ± 129.72 (0.173) | 45.02 ± 86.51 (0.076) | 27.11 ± 51.18 (0.078) | 38.73 ± 68.87 (0.071) | -1.34 ± 11.36 (0.522) |

The data is expressed as fold change in the mRNA levels in the conjunctival imprint cells post-VR surgery compared to the pre-surgery. P-value is a comparison between pre and post vitreo-retinal surgery,

*p<0.05 for each of the corresponding parameter.

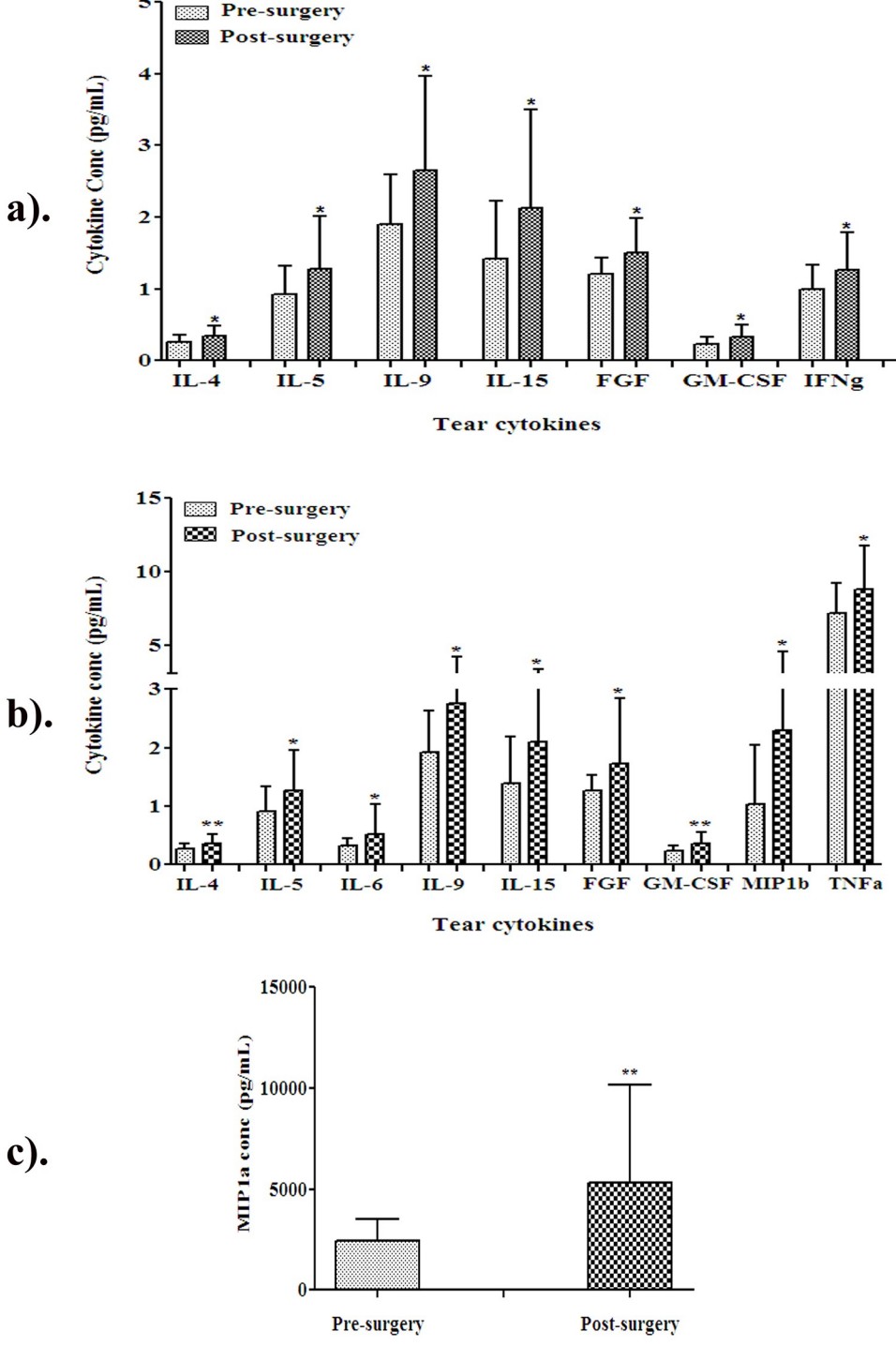

**Fig 3. Tear cytokines profiling in diabetic and age groups after VR-surgery compared to pre-surgery. a)** Significantly altered tear cytokines profile after VR surgery in DM cases (n = 24). **b)** Changes in the tear cytokines profile in sub-group of patients $\geq$ 51 years (n = 29). **c)** MIP1a tear cytokine shows higher level. The cytokine levels were expressed as pg/ml. *p<0.05, **p < 0.01. P-value is comparison between pre and post VR surgery.

**Table 5. Biological process of altered tear proteins in the top ten modules based on string analysis.**

| S. No | Biological process | No of Proteins from Modules | Nodes | FDR $q$ value |
|---|---|---|---|---|
| 1 | Cellular response to stimulus | 12 | IL2, IL4, IL5, IL6, IL15, CCL4, CSF2, IFNg, PDGFb, FGF1, MUC5AC, AQP4 | 0.015 |
| 2 | Regulation of multicellular organismal process | 10 | IL1RN, IL2, IL4, IL5, IL6, IL9, IL15, CSF2, IFNg, PDGFb | 0.0004 |
| 3 | Defense response | 9 | IL1RN, IL5, IL6, IL9, IL15, CCL4, PDGFb, FGF1, AQP4 | 0.002 |
| 4 | Response to organic substance | 9 | IL1RN, IL4, IL5, IL6, IL15,CSF2, PDGFb, FGF1, AQP4 | 0.003 |
| 5 | Cellular response to chemical stimulus | 9 | IL4, IL5, IL6, IL15, CCL4, CSF2, PDGFb, FGF1, AQP4 | 0.002 |
| 6 | Cell–cell signaling | 7 | IL1RN, IL2, IL6, IL15, CCL4, PDGFb, FGF1 | 0.0006 |
| 7 | Regulation of immune system process | 7 | IL5, IL6, IL15, CCL4, IFNg, PDGFb, FGF1 | 0.003 |
| 8 | Regulation of response to stress | 7 | IL1RN, IL2, IL4, IL6, IL15, PDGFb, IFNg | 0.004 |
| 9 | Homeostatic process | 6 | IL1RN, IL2, IFNg, AQP4, AQP5, MUC4 | 0.019 |
| 10 | Inflammatory response | 6 | IL1RN, IL5, IL6, IL9, IL15, CCL4 | 0.0002 |

FDR: False discovery Rate.

MIP1b(CCL4) and TNFa; the mucins namely MUC5AC, MUC4, MUC16; and the aquaporins namely, AQP4 and AQP5.

Functional signaling pathway analysis with the number of nodes as 18, number of edges as 70, average node degree as 7.78; average local clustering coefficient as 0.846, expected number of edges as 10 and Protein–Protein Interaction enrichment p-value as<1.0e-16. Pathway analysis with false discovery rate (FDR) $q$ value <0.05 was considered as statistically significant. The involvement of each module in different biological process was assessed and the top 10 modules are listed in Table 5. Accordingly, the molecular changes observed in the table are possibly a protective defense response to the stimulus of surgical intervention.

## Discussion

There is a paucity of reports regarding prevalence and severity of dry eye after VR surgeries. In this study, dry eye was evaluated at molecular level based on alteration in tear fluid cytokines, mucins and aquaporins that were probed before and following VR surgery. No significant change was observed in the clinical signs or symptoms of DES evaluated after 8 weeks of VR surgery that included tear osmolarity. DES is accompanied by increased tear osmolarity and inflammation of the ocular surface. Osmolarity values greater than 308 mOsm/L are a sensitive indicator of mild dry eye and values greater than 312 mOsm/L are indicative of moderate to severe dry eye [14]. In the present study, the tear osmolarity levels in both pre (305.8 ± 18.01 mOsm/L) and post-VR surgery cases (300 ± 11.9 mOsm/L) were in the normal range. There is clearly a need for much more sensitive indicator of mild dry eye. Our study showed that the conjunctival GCD was significantly reduced, and the distribution of GCD in nearly 40% cases was down-regulated in the post-operative condition in VR surgery cases. This however did not relate to the tear MUC5AC levels which were not significantly lowered possibly confounded by diabetic subjects and those aged above 50 years. Still, at gene expression levels there was an increase in the conjunctival expressions of MUC5AC, MUC4 and MUC16, [possibly a protective response. The levels of certain mucin molecules are associated with specific ocular surface states. For example, tear MUC5AC was reduced in symptomatic contact lens wearers, and MUC4 in temporal lid parallel conjunctival folds (conjunctivochalasis) and lid wiper epitheliopathy [15]. Sjogren syndrome patients exhibited increased soluble MUC16 [16]. The membrane associated MUC16 and the mucin-associated T-antigen carbohydrate were associated

with ocular surface epithelial protection [17]. In this study, post-VR surgery we see an up-regulation of the mucin gene expressions studied along with GC loss. Homeostatic control of GC was reported by De Paiva et al [18]. Changes in the GCD and mucin gene expressions act as early markers of DES and whether they are characteristic of the stimulus needs further studies. The study showed an increase of IFNg levels in the tear post VR surgery by 2-fold in 64 percent cases. Destruction and/or dysfunction of goblet cells may cause a loss of immunosuppressive properties and further enhance chronic inflammation by loss of their negative feedback to dendritic cells [19]. Contreras-Ruiz and Masli [19] reported that goblet cells have more functions and that they also produce more substances apart from mucins. Conjunctival goblet cells produce secretory mucins that bind the water at the ocular surface. Goblet cell contains small or large volumes of mucins depending on their functional state and the conjunctiva is innervated similar to the aqueous lacrimal glands [20]. Our studies showed that AQP4 gene expression was significantly up-regulated but AQP5 gene expression was down regulated. While AQP5 is characteristic of lacrimal gland/corneal/conjunctival expression, AQP4 is reportedly secreted in the retinal Muller cells, optic nerve apart from ciliary epithelium [21]. Lowered AQP5 and increased AQP4 was reported in the lacrimal gland of pregnant rabbits with DES [22].

During pars plana vitrectomy, conjunctiva is traumatized especially if scleral depression is used for surgery in the region of the vitreous base. Posterior segment surgery can lead to morphological alterations of the anterior segments as conjunctiva and distributional changes in ocular mucins, which may cause dry eye symptoms [3,4]. The effects of dry eye on the structure and function of ocular mucins are still unclear and are even contradictory [23]. In ocular surface AQP4 and AQP5 function primarily as water selective transporters and AQP5, in particular, is associated with tear formation by the lacrimal gland [24]. Bhattacharya et al., [25] showed that AQP4 and AQP5 levels are up-regulated from 1 to 3 months, and returned to near baseline by 4 months after excision of lacrimal glad in mice model. AQP4 and AQP5, among other conjunctival water transporters, appear to be involved in restoring the ocular surface fluids. In this study we observed lowering of conjunctival AQ4 and AQ5 gene expression post-VR surgery at 8 weeks which is 2 weeks after stopping the topicals. Studies are warranted to further check the protein expression of the AQPs. Remarkably, not all AQPs function as a water permeable channel. However, based on their vital role in maintaining ocular function and their roles in disease, AQPs represent potential targets for future therapeutic development [26].

A disturbance of postoperative inflammation response after laser treatment is reported after 30 days [27]. After 7 days of femtosecond laser-assisted cataract surgery rise in proinflammatory cytokines in the aqueous humor was observed [28]. Our previous study report on tear cytokine in chronic SJS cases who underwent mucous membrane showed characteristic changes post-MMG in response to the mucous membrane graft. [9]. Surgical trauma does alter the immune homeostasis with changes in pro- as well as anti-inflammatory cytokines and seems to be important in planning the therapeutic strategies, post-surgery [29]. This cellular response may vary with tissue type though there can be overlap in some of the cytokine/chemokine/growth factors expression. It can also vary with patient status such as immunocompromised patient [30]. Thus, the type of tissue and surgical insult as well as sampling time post-surgery determine the cytokine status apart from the age/sex and diabetic status of the patient.

## Dry eye in DM cases undergoing VR surgery

In DM cases, conjunctival GCD was significantly decreased after VR surgery unlike the non-diabetic cases. Diabetic patients are more prone to suffering from dry eye than normal subjects

[31]. Significant associations have been identified between diabetic retinopathy and DES [32]. Diabetes mellitus causes corneal and conjunctival epithelial damage, inducing reduction of the number of goblet cells; it reduces mucin production and the hydrophilic nature of the ocular surface leading to tear film instability [33]. Though not supported by clinical signs and symptoms, lowered GCD and lowered MUC5AC indicates dry eye in DM cases who underwent VR surgery especially in the vitreous hemorrhage, non-suture and conjunctival non-opening group. This is further supported by the fact the maximal cytokine response was also seen in these groups (S2 Table). The study showed increase in MUC5AC and MUC4 in non-DM cases possibly a protective response, while DM cases fail considerably to elicit this protective response. There are no studies that has evaluated dry eye in DM cases undergoing VR surgeries.

Ryan et al [34] reported conjunctival GCD decreased after photorefractive keratectomy or laser in situ keratomileusis (LASIK) surgery. Kato et al., reported decreased conjunctival GCD after cataract surgery [35]. Many studies have reported significantly decreased MUC5AC secretion by goblet cells in patients with dry eye [36,37]. Mucins play an important role by contributing to the wettability, lubrication, and barrier function on the ocular surface. In dry eye, the expression of both secreted and membrane-associated mucins is reportedly decreased, thereby functionally affecting the ocular surface [20].

In the present study, AQP5 expression was significantly lowered characteristically in DM cases as well as in cases aged $\geq$ 51 years post -VR surgery compared to pre-surgery. AQP5 highly expressed in lacrimal and salivary gland was reported to be lowered in a rabbit model of dry eye and the interaction between MUC5AC and AQP5 was pointed out [38]. However, in Sjogrens syndrome the tear AQP5 protein is reported to be increased. Studies are required to further check the aquaporins in various types of DES.

The study also showed seven of the cytokines that were significantly increased after VR surgery in DM cases which included IL4, IL5, IL9, FGF-basic that are anti-inflammatory and IL15, GMCSF, IFNg which are pro-inflammatory cytokines. However, association between specific cytokine change and the type of VR surgical insult needs to be established in a larger study cohort. In non-DM cases, there was no significant difference in the cytokine levels in pre and post-surgery cases. In DM cases, the significantly up-regulated tear cytokines revealed increase in IL15 (77% cases), followed by IL9 (73%), IL4 (68%), IL5 (68%), FGF Basic (68%), GMCSF (63%), and IFNg (59%) cases. Increase in IL-1$\beta$ and TNF-$\alpha$ levels in the basal epithelium of diabetic patients with dry eye has been reported [39]. There are no previous reports on a detailed cytokine profile in tears obtained from pre and post-VR surgeries in diabetic patients. Altered cytokines may have implication in the mucin expression as well. Altered mucin gene expression regulated by pro-inflammatory cytokines has been reported [40].

### Dry eye based on age and gender cases undergoing VR surgery

DES classified based on gender showed that in female cases the GCD was significantly down-regulated with no other alterations in cytokines, mucin and AQPs. In contrast, development of post-vitrectomy dry eye not dependent on age and gender was reported by Suresh et al [41]. Several studies report a higher prevalence of DES in females [42–44]. At molecular levels males are reported to be more commonly affected than females as evaluated in age more than 65 years [45]. DES in patient's $\geq$51 years of age post-VR surgery showed significant decrease in GCD. Age $\geq$51 years showed significantly altered cytokine response. AQP4 gene expression level was observed to be up-regulated after VR surgeries. Thus, age and female gender influenced the occurrence of dry eye in VR cases post-surgery. It is therefore observed that patients undergoing VR surgeries aged above 51 years, especially females and those who are diabetic

are prone to dry eye. Follow up studies are required to see if the DES progresses or self resolves. The reported prevalence of DES in diabetics over 65 years of age is 15–33% and increases with age. Dry eye is 50% more common in women than in men [46].

### Protein-protein interaction analysis

Eighteen significant molecules imported on string database was found to play direct or indirect interaction within themselves and with other biological molecules. Based on this interaction, the molecules are shown to be involved in potent biological pathway centering around cellular response to stimulus as part of regulatory mechanism towards homoeostasis. Previous reports in tear fluid proteome analysis in dry eye patients have shown similar findings [13, 47].

## Conclusion

To summarize, we observed that after vitreo-retinal surgical insult, there was no significant change in clinical signs or symptoms of dry eye syndrome. Though no significant change in the protein level of MUC5AC was seen, lowered goblet cell density was seen post-vitreoretinal surgery indicating dry eye. The aquaporins which maintain hydration and/or cellular homeostasis in ocular tissues were found to be altered especially with lowering of AQP5 indicating functional changes in the goblet cells as well. Tear cytokines were altered indicating a defensive response. Increase in MUC4 and MUC16 gene expression observed indicate a protective pro-liferative response in the conjunctival epithelium. These might serve as early markers of dry eye disease. Evaluation of goblet cell in terms of its functional loss of mucilage formation needs further attention.

Thus, in the absence of symptomatic dry eye postvitreo-retinal surgery, conjunctival changes in the form of reduced goblet cell density, altered MUC4, MUC16, AQP5 and cytokines are suggestive of dry eye at a molecular level and based on goblet cell density. More importantly, the study reveals that patients undergoing VR surgeries who are aged above 51 years, especially females and those who are diabetic are prone to dry eye development as assessed by these molecular changes in the ocular surface along with lowering of goblet cell density.

Smaller sample size in sub-group analysis and lack of follow-up status of the molecular markers is the limitation in the study. Sensitive methods of tear mucin protein changes in larger sample size are warranted. Validation of the molecular markers of the dry eye in a larger cohort that includes follow-up in cases undergoing vitreo-retinal surgery beyond 8 weeks needs to be done in further studies.

## Supporting information

**S1 Fig. Distribution analysis of conjunctival goblet cell density showing the median, in non-DM and DM cases.** Non-DM (n = 20) and DM cases (n = 24). *p<0.05 is the comparison between pre and post VR surgery; # p < 0.05 is the comparison between post-VR surgery in non-DM vs DM.
(JPG)

**S2 Fig. Distribution analysis with median of tear cytokines concentration in DM cases of pre and post VR surgery.** The bar graph depicting percentage of cases with significantly up-regulated tear cytokines after VR surgery. (n = 24); *p<0.05 is the comparison between pre and post VR surgery.
(JPG)

**S1 Table. Tear cytokines profile pre and post vitreo-retinal surgery.** p-value is a comparison between pre and the corresponding post-VR surgery tear cytokine. $^*p<0.05$ and $^{**}p<0.01$. Statistically significant p-values are given in bold. Sample size = 36.
(DOCX)

**S2 Table. Cytokines profiling based on VR surgery.** p-value is a comparison between pre and the corresponding post-VR surgery tear cytokine. $^*p<0.05$, $^{**}p<0.01$ and $^{***}p<0.001$. Statistically significant p-values are given in bold. Sample size = 36.
(DOCX)

## Author Contributions

**Conceptualization:** Ratra Vineet, Ratra Dhanashree, Narayanasamy Angayarkanni.

**Data curation:** Ramalingam Mani, P. S. Shobha, Saravanan Thilagavathi, Natarajan Viswanathan, Ratra Dhanashree.

**Formal analysis:** Ramalingam Mani, P. S. Shobha, Saravanan Thilagavathi, Natarajan Viswanathan, Ratra Dhanashree, Narayanasamy Angayarkanni.

**Funding acquisition:** Ratra Dhanashree.

**Investigation:** Ramalingam Mani, P. S. Shobha, Saravanan Thilagavathi, Ratra Dhanashree, Narayanasamy Angayarkanni.

**Methodology:** Ramalingam Mani, Padmanabhan Prema, Ratra Dhanashree, Narayanasamy Angayarkanni.

**Project administration:** Ratra Dhanashree, Narayanasamy Angayarkanni.

**Resources:** Ratra Dhanashree, Narayanasamy Angayarkanni.

**Software:** Ramalingam Mani, Natarajan Viswanathan.

**Supervision:** Padmanabhan Prema, Ratra Vineet, Ratra Dhanashree, Narayanasamy Angayarkanni.

**Validation:** Ramalingam Mani, Ratra Dhanashree, Narayanasamy Angayarkanni.

**Visualization:** Ramalingam Mani, Ratra Dhanashree, Narayanasamy Angayarkanni.

**Writing – original draft:** Ramalingam Mani, Ratra Dhanashree, Narayanasamy Angayarkanni.

**Writing – review & editing:** Ramalingam Mani, Ratra Vineet, Ratra Dhanashree, Narayanasamy Angayarkanni.

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
