## [Decision Letter · Decision Letter 0]

31 Jan 2020

PONE-D-19-34984

Altered mucins and aquaporins indicate dry eye outcome in patients undergoing Vitreo-retinal surgery

PLOS ONE

Dear Dr. Angayarkanni ,

Thank you for submitting your manuscript to PLOS ONE. After careful consideration, we feel that it has merit but does not fully meet PLOS ONE’s publication criteria as it currently stands. Therefore, we invite you to submit a revised version of the manuscript that addresses the points raised during the review process. Please note it is required that you address all the issues raised by the reviewers.

We would appreciate receiving your revised manuscript by May 1st, 2020. To enhance the reproducibility of your results, we recommend that if applicable you deposit your laboratory protocols in protocols.io, where a protocol can be assigned its own identifier (DOI) such that it can be cited independently in the future. For instructions see: http://journals.plos.org/plosone/s/submission-guidelines#loc-laboratory-protocols

We look forward to receiving your revised manuscript.

Kind regards,

Deepak Shukla

Academic Editor

PLOS ONE

Journal Requirements:

"Novartis Healthcare Private Limited"       

3. In your data availability statement you write, "All relevant data are within the paper and its Supporting Information files." Please ensure you have provided the individual data points used to create the figures and determine means, medians and variance measures presented in the results, tables and figures (http://journals.plos.org/plosone/s/data-availability#loc-faqs-for-data-policy). If these data cannot be publicly deposited or included in the supporting information, e.g. due to patient privacy or ownership by a third party, explain why and explain how researchers may access them.

4. Thank you for your ethics statement:

'The study was conducted as a hospital based prospective observational study after institution ethics committee approval. Written and informed consent of the patients was obtained.'

Reviewers' comments:

Reviewer's Responses to Questions

**Comments to the Author**

1. Is the manuscript technically sound, and do the data support the conclusions?

Reviewer #1: Partly

Reviewer #2: Partly

2. Has the statistical analysis been performed appropriately and rigorously? 

Reviewer #1: Yes

Reviewer #2: Yes

3. Have the authors made all data underlying the findings in their manuscript fully available?

Reviewer #1: Yes

Reviewer #2: Yes

4. Is the manuscript presented in an intelligible fashion and written in standard English?

Reviewer #1: Yes

Reviewer #2: No

5. Review Comments to the Author

Reviewer #1: Dry eye disease (DED) is clearly a multifactorial condition yet it is wholly a clinical diagnosis The tolerance of a given individual to the studied molecular changes may or may not be harbingers of future DED. The incorporation of DM patients, those undergoing retina detachment procedures and the newer trans conjunctival vitrectomy procedures in a single opus is confusing. Each cohort should be treated separately. The use of topical steroids postoperatively was not discussed. Were there significant variations in administration or was a single protocol utilized.

Reviewer #2: This is an interesting article relating the occurrence of Dry eye syndrome after vitrectomy. The methods performed in this manuscript are accurate and reasonable. All the conditions for inclusion and exclusion are done correctly. However the conclusions made in this article seem a little over-reaching. While it is true that multiple case studies have shown the occurrence of DES with vitrectomy, the results from this study do not seem to be as conclusive as one would like to see.

The pros of this manuscript are the recommendations made to Diabetic patients. It is also very interesting to see that mucin secretion and GCD go down significantly after the surgical procedure. The cons include low and under-explanation of the results and almost complete absence of discussion of the results. I would recommend the authors to reconsider how they portray the results better in an intelligible fashion. Also I do not understand the purpose of Figure 4. It is neither very informative nor self explanatory nor the authors have even attempted to explain whats going on in this figure. I would infact recommend the complete removal of this figure unless a better explanation is made.

Another question I had about the article is whether cytokines are elevated after any ocular surgery? Does any surgery cause cytokine increase or does this particular surgery casue it? May be some information regarding this in the discussion section can carve out a better explanation about the particular vitrectomy as a responsible factor for DES.

The authors may also consider explaining the acronyms for the genes better in the manuscript. This would increase the readability of the paper.

6. PLOS authors have the option to publish the peer review history of their article (what does this mean?). If published, this will include your full peer review and any attached files.

Reviewer #1: No

Reviewer #2: No

---

## [Author Response · Author response to Decision Letter 0]

7 Apr 2020

Respond to reviewers documents to be attached at attach files.

---

## [Decision Letter · Decision Letter 1]

7 May 2020

Altered mucins and aquaporins indicate dry eye outcome in patients undergoing Vitreo-retinal surgery

PONE-D-19-34984R1

Dear Dr. Angayarkanni,

We are pleased to inform you that your manuscript has been judged scientifically suitable for publication and will be formally accepted for publication once it complies with all outstanding technical requirements.

With kind regards,

Deepak Shukla

Academic Editor

PLOS ONE

Additional Editor Comments (optional):

Reviewers' comments:

Reviewer's Responses to Questions

**Comments to the Author**

1. If the authors have adequately addressed your comments raised in a previous round of review and you feel that this manuscript is now acceptable for publication, you may indicate that here to bypass the “Comments to the Author” section, enter your conflict of interest statement in the “Confidential to Editor” section, and submit your "Accept" recommendation.

Reviewer #1: All comments have been addressed

Reviewer #2: All comments have been addressed

2. Is the manuscript technically sound, and do the data support the conclusions?

Reviewer #1: Yes

Reviewer #2: Yes

3. Has the statistical analysis been performed appropriately and rigorously? 

Reviewer #1: Yes

Reviewer #2: Yes

4. Have the authors made all data underlying the findings in their manuscript fully available?

Reviewer #1: Yes

Reviewer #2: Yes

5. Is the manuscript presented in an intelligible fashion and written in standard English?

Reviewer #1: Yes

Reviewer #2: Yes

6. Review Comments to the Author

Reviewer #1: (No Response)

Reviewer #2: All concerns have been addressed. This article is now suitable for publication in the current journal.

7. PLOS authors have the option to publish the peer review history of their article (what does this mean?). If published, this will include your full peer review and any attached files.

Reviewer #1: No

Reviewer #2: No

---

## [Editor Report · Acceptance letter]

12 May 2020

PONE-D-19-34984R1 

Altered mucins and aquaporins indicate dry eye outcome in patients undergoing Vitreo-retinal surgery 

Dear Dr. Angayarkanni:

I am pleased to inform you that your manuscript has been deemed suitable for publication in PLOS ONE. Congratulations! Your manuscript is now with our production department. 

With kind regards,

on behalf of

Prof. Deepak Shukla 

Academic Editor

PLOS ONE